# Oxygen-Deficient Stannic Oxide/Graphene for Ultrahigh-Performance Supercapacitors and Gas Sensors

**DOI:** 10.3390/nano11020372

**Published:** 2021-02-02

**Authors:** Liyang Lin, Susu Chen, Tao Deng, Wen Zeng

**Affiliations:** 1School of Aeronautics, Chongqing Jiaotong University, Chongqing 400074, China; d82t722@cqjtu.edu.cn; 2The Green Aerotechnics Research Institute, Chongqing Jiaotong University, Chongqing 400714, China; 3College of Aerospace Engineering, Chongqing University, Chongqing 400044, China; 20203101007g@cqu.edu.cn; 4College of Materials Science and Engineering, Chongqing University, Chongqing 400044, China; wenzeng@cqu.edu.cn

**Keywords:** oxygen vacancy, r-SnO_2_/GN, supercapacitors, gas sensors

## Abstract

The metal oxides/graphene nanocomposites have great application prospects in the fields of electrochemical energy storage and gas sensing detection. However, rational synthesis of such materials with good conductivity and electrochemical activity is the topical challenge for high-performance devices. Here, SnO_2_/graphene nanocomposite is taken as a typical example and develops a universal synthesis method that overcome these challenges and prepares the oxygen-deficient SnO_2_ hollow nanospheres/graphene (r-SnO_2_/GN) nanocomposite with excellent performance for supercapacitors and gas sensors. The electrode r-SnO_2_/GN exhibits specific capacitance of 947.4 F g^−1^ at a current density of 2 mA cm^−2^ and of 640.0 F g^−1^ even at 20 mA cm^−2^, showing remarkable rate capability. For gas-sensing application, the sensor r-SnO_2_/GN showed good sensitivity (~13.8 under 500 ppm) and short response/recovering time toward methane gas. These performance features make r-SnO_2_/GN nanocomposite a promising candidate for high-performance energy storage devices and gas sensors.

## 1. Introduction

With the development of society and the improvement of the level of science and technology, people are increasingly improving their own quality of life. In particular, terms such as health, safety, and convenience frequently appear in the public’s field of vision [1,2,3]. Therefore, environmental safety monitoring, human safety monitoring and green energy are facing huge business opportunities and challenges, including electrochemical energy storage devices [4,5,6], gas sensor devices [7,8,9], pressure sensor devices [10,11,12], and so on. This also puts forward higher requirements for the comprehensive performance of the material.

Whether in electrochemical energy storage devices or gas sensors, metal oxides such as NiO [13,14], ZnO [15,16], Co_3_O_4_ [17,18], SnO_2_ [19,20], etc. play a very important role in their material design and composition. However, these metal oxides themselves have poor electrical conductivity. Moreover, it is inevitable to introduce polymers used as binders when assembling devices. Even a very small amount will cause the electron or ion transport channels to be cut off and significantly reduce the performance. In recent years, a large number of researches have been conducted on the use of metal oxide nanomaterials as the matrix and the selection of suitable carbon materials for doping to form metal oxide/carbon nanocomposites [21,22,23]. The synergistic effect due to compounding can effectively improve the overall characteristics, especially the conductivity. However, carbon nanomaterials such as graphene are prone to agglomeration due to their own characteristics, so the performance improvement effect is not obvious, which greatly limits the advantages of this method [24]. Fortunately, recent studies have shown that changing the surface properties of metal oxides by creating oxygen vacancies can effectively improve performance, which provides a new idea for the synthesis of metal oxide/carbon composites [25,26,27,28,29].

SnO_2_ is an important semiconductor material for gas sensors and supercapacitors. At present, the main method to improve the gas-sensing performance of SnO_2_ materials is to increase the crystal defects by creating oxygen vacancies or doping impurities, thereby increasing its carrier density [30]. Bunpang et al. synthesized flame-spray-made Cr-doped SnO_2_ nanoparticles to improve its gas-sensing performance toward CH_4_ [31]. Bonu et al. studied the influence of in-plane and bridging oxygen vacancies of SnO_2_ nanostructures on CH_4_ sensing at low operating temperatures [32]. Kooti et al. prepared SnO_2_ nanorods–nanoporous graphene hybrids and speculated that the substantial enhancement in the gas detection of SnO_2_ NRs-NPG nanohybrid should be attributed to its larger specific surface area, the nanoporous graphene as a good conductor, and the synergistic effect [33]. In contrast, SnO_2_ as a supercapacitor electrode material has not achieved satisfactory electrochemical performance, and even the use of carbon materials for compounding cannot effectively improve its specific capacitance and rate performance. Notably, the effect of oxygen vacancies on the electrochemical property of SnO_2_ has never been systematically investigated.

In this work, graphene modified oxygen-deficient SnO_2_ nanocomposite (r-SnO_2_/GN) was successfully synthesized via the solvothermal method and annealing treatment. Different from the reported SnO_2_ nanostructures [34,35,36,37], this paper synthesized SnO_2_ hollow spheres with large specific surface area and stable structure by means of concentrated acid and organic solution at appropriate reaction temperature and reaction time. At the same time, chemical treatment and heat treatment were reasonably combined in the introduction of oxygen vacancies, which not only achieved the deep reduction of GO but also introduced a large number of oxygen vacancies in SnO_2_ under the premise of ensuring the phase stability. A series of characterizations such as XRD, Raman, SEM, TEM, BET, and XPS were conducted to reveal its stable hybrid microstructure, perfect composition and abundant oxygen vacancies before the assembling of r-SnO_2_/GN nanomaterials. Both as the electrode material of supercapacitors and gas-sensing material, it shows excellent performance.

## 2. Materials and Methods

### 2.1. Material Preparation

Preparation of GO: GO nanomaterials were synthesized from natural graphite by the popularly used Hummers method [38].

Synthesis of Hollow SnO_2_ nanospheres: 0.190 g of SnCl_4_ (Reagent. Chengdu Kelong Co. Ltd., Chengdu, China) was fully dissolved in a mixed solution of 5 mL of deionized water and 50 mL of absolute ethanol, and then 0.5 mL of 37% HCl (Reagent. Chongqing Chuandong Chemical Co., Ltd., Chongqing, China) was added. The above solution was ultrasonically stirred for 30 min and then transferred into 100 mL Teflon-lined stainless steel autoclave at 200 °C for 24 h. The white precursor was centrifuged and washed with deionized water and absolute ethanol three times. The powder was annealed further at 400 °C in air for 2 h.

Preparation of r-SnO_2_/GN nanocomposite: As-obtained SnO_2_ (60 mg) and GO (10 mg) were dispersed in 40 mL of deionized water for 1.5 h using an ultrasonic reactor, then stirred with a planetary mixer for 5 min. The mixture was immersed in NaBH_4_ solution (10 mg mL^−1^) for 10 h. After centrifugal cleaning and drying, the black powder was then placed in a tube furnace filled with nitrogen and calcined at 500 °C for 3 h.

### 2.2. Material Characterization

Morphologies and nanostructures were characterized with FE-SEM (JEOL, JSM-7800F, Tokyo, Japan) and TEM (FEI, TALOS F200, Waltham, MA, USA). The composition and phase of the samples were evaluated by XRD (BRUKER, D2 PHASER, Karlsruher, Germany) and XPS (Thermo Fisher Scientific, ESCALAB 250Xi, Waltham, MA, USA). Raman spectra of the samples were obtained by a micro-Raman system (HORIBA Jobin Yvon, LabRAM ARAMIS, Paris, France). Nitrogen adsorption−desorption isotherm was used to examine the specific surface area and pore structure (MicrotracBEL, BELSORP-max II, Japan).

### 2.3. Electrochemical Measurements

Preparation of working electrodes: Samples SnO_2_ and r-SnO_2_/GN were separately blended with carbon black and Polyvinylidene Fluoride (PVDF) at a mass ratio of 7:2:1 in NMP. The above slurry was smearing immediately onto the as-cleaned Ni foam substrate and then dried at 50 °C in vacuum for 12 h (the coating area was controlled to be 1 cm × 1 cm).

Measurement conditions: Pt foil, saturated calomel electrode (SCE), and 3.0 M KOH solution was used as counter electrode, reference electrode, and electrolyte, respectively. The specific capacitance was calculated by the common formula *C_m_* = *I*Δ*t/m*Δ*U* in the discharge measurements, where *I* is discharge current density, Δ*t* is the discharging time, *m* is the mass of the active materials, and Δ*U* is the width of the potential window [39]. The frequency of the electrochemical impedance spectra was collected from 0.01 Hz to 100 kHz. The mass loading of the active material for SnO_2_ and r-SnO_2_/GN electrodes was about 3.6 mg cm^−2^ and 3.5 mg cm^−2^, respectively.

### 2.4. Gas-Sensing Measurements

Preparation of gas sensors: Typically, samples SnO_2_ and r-SnO_2_/GN were separately dispersed into distilled water using an ultrasonic cleaner. The obtained slurries were uniformly coated onto the surface of specific alumina ceramic tubes and then dried in an oven [40].

Measurement conditions: Gas-sensing properties were measured by a chemical gas sensor-8 intelligent gas-sensing analysis system (Beijing Elite Tech. Co., Ltd., Beijing, China). The response (*S*) was defined as *S* = *R*_a_/*R*_g_, where *R*_a_ and *R*_g_ were initial and real-time resistance, respectively.

## 3. Results and Discussion

Figure 1a shows XRD patterns of two samples SnO_2_ (I) and r-SnO_2_/GN (II). For sample I, it can been seen that six characteristic peaks at 26.6°, 33.9°, 37.9°, 51.8°, 54.8°, and 65.9° correspond to (110), (101), (200), (211), (220), and (301) planes of the tetragonal phase of SnO_2_ (JCPDS 41-1445). All these peaks are sharp and clean without any impurity peaks. It can also be seen from the figure that the diffraction pattern of r-SnO_2_/GN is very similar to that of SnO_2_. In addition, there is no obvious characteristic peak at 25.1° corresponding to GN. Raman spectra for SnO_2_ (I) and r-SnO_2_/GN (II) are shown in Figure 1b. Compared with pure SnO_2_, r-SnO_2_/GN nanocomposite exhibits intense peaks at 1360 and 1575 cm^−1^, corresponding to the D and G bands of GO. As we know, the ratio of *I*_D_/*I*_G_ can be used to measure the redox degree of the graphite oxide. The value of *I*_D_/*I*_G_ for the r-SnO_2_/GN is about 1.16 higher than that of GO (about 0.98), indicating the successful reduction of GO to rGO after the chemical and heat treatments. In addition, there is a wide band with a peak near 570 cm^−1^ corresponding to SnO_2_, which reveals the successful formation of r-SnO_2_/GN nanocomposite. In addition to Raman spectra, the following SEM, TEM, and XPS results will provide more evidence. Therefore, it can be inferred that the surface of SnO_2_ is coated with disordered stacking but uniform dispersing of GN [41].

SEM images of two samples are shown in Figure 2. At low magnification (Figure 2a), it can be found that the SnO_2_ powder is composed of spheres with uniform size and rough surface. The diameter of each sphere is about 300 nm. Moreover, it can be observed that the rough spherical surface is composed of countless tiny particles under higher magnification (Figure 2b). Figure 2c,d shows the morphologies of sample r-SnO_2_/GN. Even after a long time immersion in NaBH_4_ solution and another heat treatment, the SnO_2_ nanostructure did not break and still appeared as a sphere with a rough surface. The difference is that the surface is obviously wrapped with gauze-like nanomaterials. Compared with the former, the latter has a small amount and no obvious agglomeration, which verifies the typical GN morphology inferred from XRD results. Besides, EDS mapping including Sn, O, and C elements is also collected and shown in Figure 2e. It can be seen that all these elements are uniformly distributed comparing to the original SEM image, indicating the successful deposition of GN on the surface of r-SnO_2_ spheres to form a stable composite.

The nanostructures of SnO_2_ and r-SnO_2_/GN were further investigated by conducting TEM and HRTEM imaging, as shown in Figure 3. Figure 3a,b shows a close-up of the internal morphology of sample SnO_2_, revealing its hollow structure. HRTEM image (Figure 3c) of a nanoparticle on the spherical shell exhibits clear and uniformly arranged lattice fringes. The interplanar spacing is about 0.26 nm corresponding to (101) crystal plane of the SnO_2_ phase. Herein, the formation mechanism of SnO_2_ hollow sphere is speculated. Due to the presence of concentrated hydrochloric acid, ethanol will be dehydrated to produce H_2_O. SnCl_4_ will initially hydrolyze and condense to produce primary SnO_2_ nanocrystals. Subsequently, these SnO_2_ nanocrystals aggregate and form large solid microspheres. This process is very fast and completes after a few hours. Therefore, the new solid spheres cannot crystallize well, especially for the nanoparticles inside the solid spheres. Furthermore, these internal nanoparticles with higher surface energy will dissolve under solvothermal conditions to form a core-shell structure. The dissolved Sn^4+^ will be further hydrolyzed and condensed to form SnO_2_ nanocrystals on the surface of the sphere. In addition, the temperature of 200 °C and the reaction time of 24 h are also guarantees for uniformly dispersed SnO_2_ hollow spheres. From Figure 3d, one can see sample r-SnO_2_/GN maintaining the hollow structure and successfully being covered by gauze-like nanosheets. HRTEM image focusing on the interface (red rectangle in Figure 3e) has been shown in Figure 3f. Obviously, in addition to the lattice fringes corresponding to SnO_2_, another one with interplanar spacing of 0.35 nm can be determined, which is indexed to (002) crystal plane of the hexagonal phase of C. Moreover, the few-layer GN are curled up and tightly combined with SnO_2_ grains.

Pore size distribution curve and N_2_ adsorption–desorption isotherm of two samples are shown in Figure 4. According to the experimental data, the specific surface area of sample SnO_2_ is 43.58 m^2^ g^−1^ and the average pore size is about 3.86 nm. By contrast, sample r-SnO_2_/GN has a larger specific surface area of 57.86 m^2^ g^−1^ and a smaller average pore size of 3.76 nm. Based on these results, it can be confirmed that the sample r-SnO_2_/GN is mesoporous and has the larger specific surface area. XPS spectrums of sample r-SnO_2_/GN and sample SnO_2_ are further obtained and shown in Figure 5. All binding energies are referenced to the C_1s_ at 284.6 eV. Figure 5a displays a survey of r-SnO_2_/GN indicating the predominant presence of Sn, C, and O elements. According to the high-resolution spectrum (Figure 5b), it can be seen that Sn 3 d spectrum of r-SnO_2_/GN exhibits two major peaks at 487.4 eV and 495.8 eV, which shifts about 0.9 eV to a higher binding energy comparing to SnO_2_ (486.5 eV and 494.9 eV). Abundant oxygen vacancies introduced in SnO_2_ crystals give a high electron-attracting effect to nearby Sn, resulting in the decrease of the electron density of nearby Sn [42,43,44]. The decrease of electron density further gives rise to the increase of the electron binding energy. The above result evidently confirms the abundant oxygen vacancies in r-SnO_2_/GN. In addition, Figure 5c shows the bonding information for C 1 s including C−C, C−O, and C=O. Obviously, the less intense residual oxygen containing functional group peaks indicates that GO has been reduced more completely [45]. Figure 5d illustrates that O 1 s spectrum of r-SnO_2_/GN exhibits two major peaks at 530.5 eV and 531.8 eV. The peak located at 530.5 eV is indexed to the lattice oxygen (O_L_) species, and the peak located at 531.8 eV is assigned to the chemisorbed oxygen (O_C_) species.

To evaluate their potential application as supercapacitors, electrochemical properties of electrodes SnO_2_ and r-SnO_2_/GN were tested. CV curves of two electrodes at a scanning rate of 50 mV s^−1^ are shown in Figure 6a for comparison. Both of them exhibit two redox peaks revealing their pseudocapacitor characteristics. However, the region surrounded by the CV curve for r-SnO_2_/GN is much broader than that for SnO_2_. At the same time, GCD curves of two electrodes at a current density of 2 mA cm^−2^ are shown in Figure 6b. As expected, the discharge time of electrode r-SnO_2_/GN is about 663.2 s much longer than that of SnO_2_ (only 133.0 s). In detail, CV curves of electrodes SnO_2_ and r-SnO_2_/GN at different voltage sweep rate are shown in Figure 6c,e, respectively. As the voltage sweep rate increases from 10 to 100 mV s^−1^, the absolute value of the redox peak (for both current density and potential) increases significantly, showing a relatively low resistance and a rapid redox reaction at the interface between the electrode and the electrolyte, especially for the electrode r-SnO_2_/GN [46]. GCD curves of two electrodes at different current densities are shown in Figure 6d,f, respectively. When the current density changes from 2 to 20 mA cm^−2^, the discharge time of the electrode r-SnO_2_/GN is 663.2, 401.7, 213.6, 100.6, and 44.8 s. Under the same condition, the electrode SnO_2_ shows 133.0, 82.5, 44.4, 15.8, and 4.7 s.

Figure 7a shows the curve chart of specific capacitance (F g^−1^). Based on the common formula mentioned in experimental part, the specific capacitance of the electrode r-SnO_2_/GN can be calculated to be 947.4, 860.8, 762.9, 715.7, and 640.0 F g^−1^ at a current density of 2, 3, 5, 10, and 20 mA cm^−2^, respectively. As a contrast, the electrode SnO_2_ only has 184.7 F g^−1^ at the low current density of 2 mA cm^−2^. As far as we know, the observed electrochemical performances are better than other SnO_2_-based materials, which have been reported before, such as the pure SnO_2_ (138 F g^−1^) [34], SnO_2_-NGO (378 F g^−1^) [35], SnO_2_/g-C_3_N_4_ (488 F g^−1^) [47], and the hollow SnO_2_ (332.7 F g^−1^) [30] (See Table 1). The cycling stability of two electrodes is shown in Figure 7b. After 1000 GCD cycles, the electrode r-SnO_2_/GN still retains about 88.2% of initial specific capacitance, better than that of the electrode SnO_2_ (72.1% remain). Moreover, EIS Nyquist plots of two electrodes (Figure 7c,d) were measured to understand their ion and electron transfer performance. EIS data were fitted according to the equivalent circuit model (the inset of Figure 7c,d). Four main parameters including the charge-transfer *R*_ct_, solution resistance *R*_s_, double-layer capacitance *C*_dl_, and Warburg resistance *W* were considered [48]. Generally, the diameter of semicircle in the high frequency range shows the *R*_ct_ and the slope of the line at low frequency region means *W*, which represent the charge transfer at the interface and the diffusion of the redox species in the electrode, respectively. After comparing the EIS data of two electrodes, it can be found that two values (*R*_ct_ and *W*) read from the curve of the electrode r-SnO_2_/GN are smaller than that of the electrode SnO_2_, indicating that that the former has better ion and electron transport capability. Several advantages of the oxygen-deficient r-SnO_2_/GN nanocomposite make them a promising candidate as the electrochemical supercapacitors. These features include: (i) Abundant oxygen vacancies facilitate easy access of electrolyte ions into the large surface area of the electrode. (ii) Highly reductive GN as facile electron transfer paths provides good interfacial contact for r-SnO_2_.

Furthermore, gas-sensing functionalities of two samples were also investigated (Figure 8). Two gas sensors toward methane gas of 500 ppm at different working temperatures were tested. Their response plots are shown in Figure 8a. It can be seen that the sensor r-SnO_2_/GN exhibits much more remarkable response value than SnO_2_ from 100 to 200 °C. The best working temperature for r-SnO_2_/GN is found around 140 °C. The above results also indicate that the sensor r-SnO_2_/GN remains an n-type semiconductor material even when doped with a certain amount of graphene. After setting the working temperature as 140 °C, the resistances of sensor r-SnO_2_/GN and SnO_2_ in air are recorded, respectively. The sensor SnO_2_ has a resistance of 14.71 MΩ in air. However, for the sensor r-SnO_2_/GN exposed to air, it exhibits stable and lower resistance (about 0.36 MΩ). From Figure 8b, a particular response curve of the sensor r-SnO_2_/GN toward methane gas at 140 °C, one can see that the response value keeps stable when the sensor is exposed in air, yet goes up rapidly once 500 ppm methane is injected into the system. It eventually maintains at ~13.8 and decreases quickly when the methane gas is out. In addition, the response values of the sensor r-SnO_2_/GN toward methane gas of various concentrations were tested and shown in Figure 8c. From 100 to 5000 ppm, the response value is about 3.2, 6.9, 13.8, 16.2, 18.1, and 20.4, respectively, revealing the law that it first increases rapidly and then tends to slow. Particularly, whether in low concentrations or high concentrations (Figure 8d), the response and recovering time is short. It means that the sensor r-SnO_2_/GN achieves good response ability. The response of r-SnO_2_/GN is also compared with other SnO_2_-based materials [36,37,49,50,51] (See Table 2). The long-term stability of the sensor base on r-SnO_2_/GN is also investigated (Figure 8e). After 6 days testing, the response nearly keeps at a constant value (13.65~13.8), demonstrating excellent long-term stability. The sensitivities of the sensor r-SnO_2_/GN toward different gases are shown in Figure 8f. In addition to a maximum response value of 13.8 for methane gas, it also has a good effect on ethanol and methanol gases. The sensor r-SnO_2_/GN possessing good gas-sensing performance should be attributed to its self-characteristics. Large specific surface area with abundant oxygen vacancies creates defect states close to the conduction band minimum, resulting in low activation energy.

## 4. Conclusions

In summary, oxygen-deficient r-SnO_2_/GN nanocomposite with large specific surface area was rationally designed and successfully fabricated. As the supercapacitor electrode, it exhibited greatly improved electrochemical performances with an ultrahigh specific capacitance of 947.4 F g^−1^ at a current density of 2 mA cm^−2^ and of 640.0 F g^−1^ even at 20 mA cm^−2^. For gas-sensing application, the sensor r-SnO_2_/GN showed good sensitivity and short response/recovering time toward methane gas. The synergistic effect of vacancy-enriched SnO_2_ and GN wrapped on its surface makes r-SnO_2_/GN a promising nanocomposite material for energy storage devices and gas sensors.

## Figures and Tables

**Figure 1 nanomaterials-11-00372-f001:**
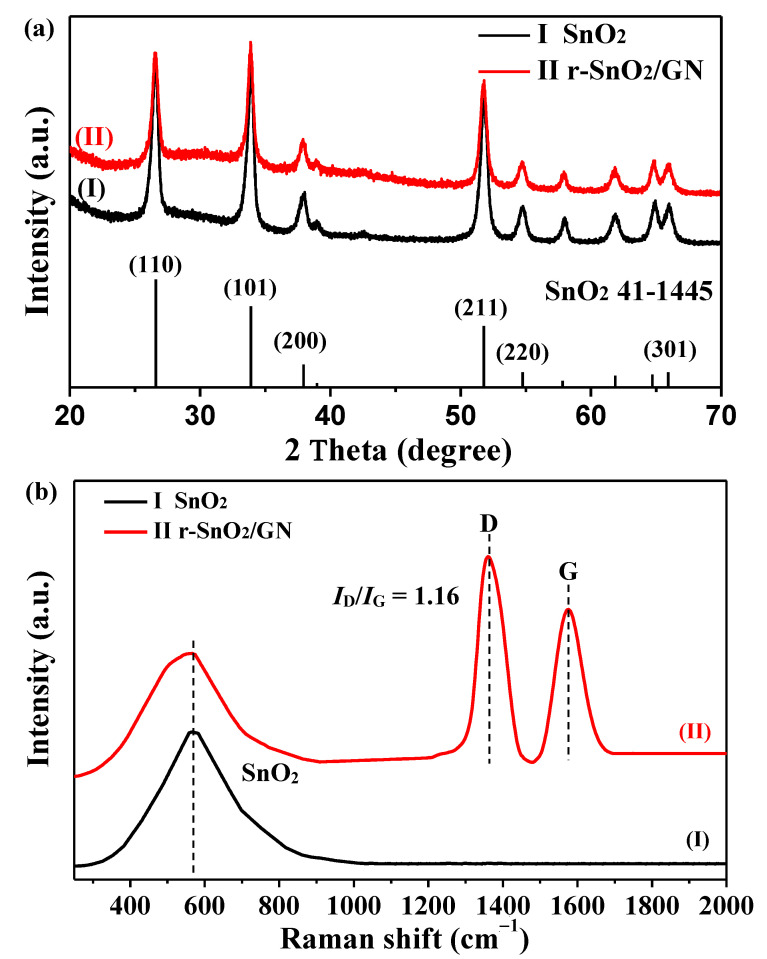
(**a**) XRD patterns of two samples SnO_2_ (I) and r-SnO_2_/GN (II); (**b**) Raman spectra of two samples SnO_2_ (I) and r-SnO_2_/GN (II).

**Figure 2 nanomaterials-11-00372-f002:**
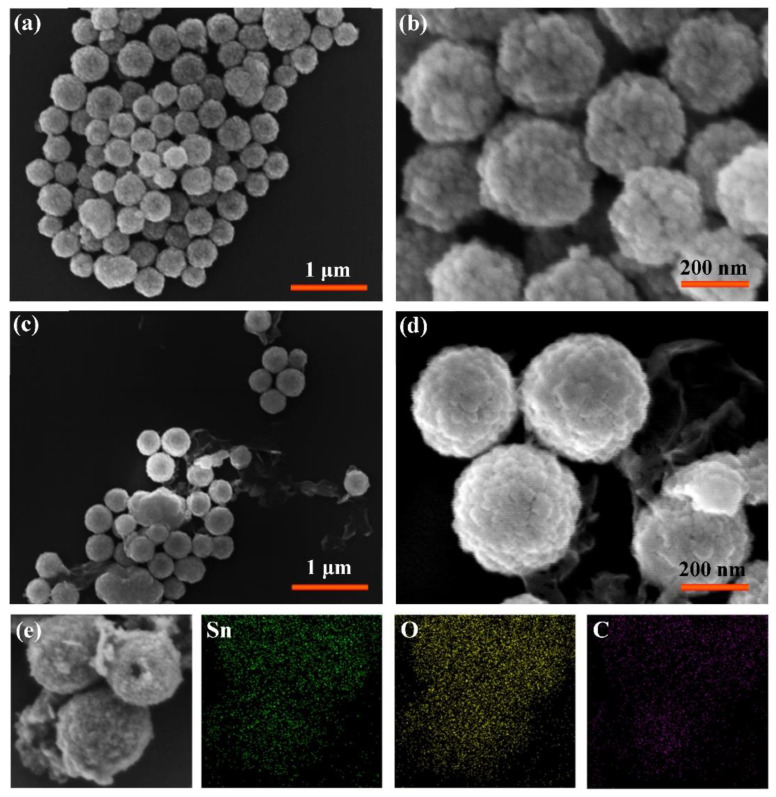
SEM images of two samples (**a**,**b**) SnO_2_ and (**c**,**d**) r-SnO_2_/GN; (**e**) EDS mapping including Sn, O, and C elements for r-SnO_2_/GN.

**Figure 3 nanomaterials-11-00372-f003:**
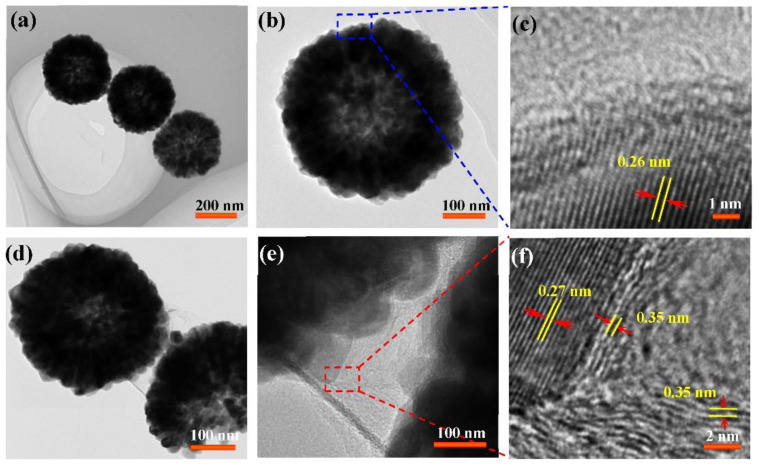
TEM images of two samples (**a**–**c**) SnO_2_ and (**d**–**f**) r-SnO_2_/GN.

**Figure 4 nanomaterials-11-00372-f004:**
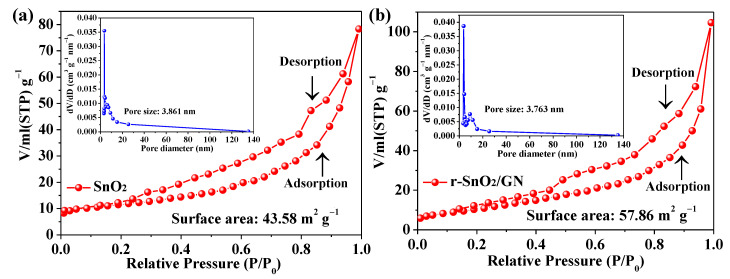
N_2_ adsorption/desorption isotherms of two samples (**a**) SnO_2_ and (**b**) r-SnO_2_/GN; the inset is the pore size distribution.

**Figure 5 nanomaterials-11-00372-f005:**
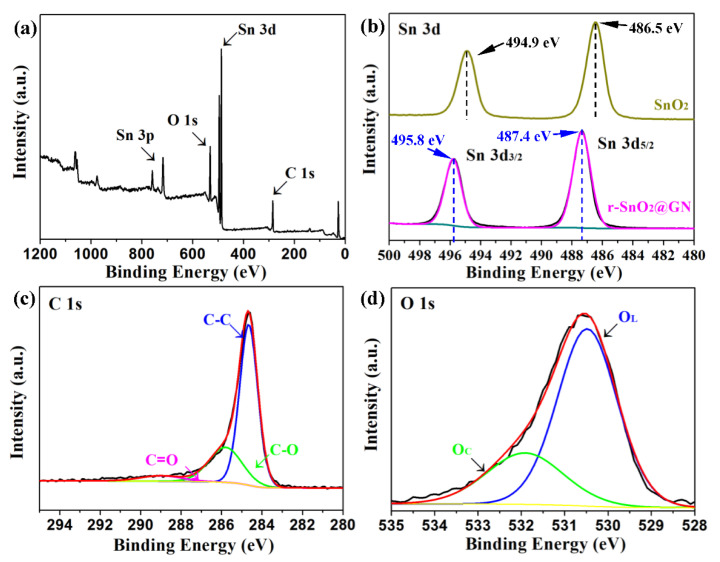
XPS of r-SnO_2_/GN and SnO_2_ (**a**) survey, (**b**) Sn 3 d, (**c**) C 1 s, and (**d**) O 1 s.

**Figure 6 nanomaterials-11-00372-f006:**
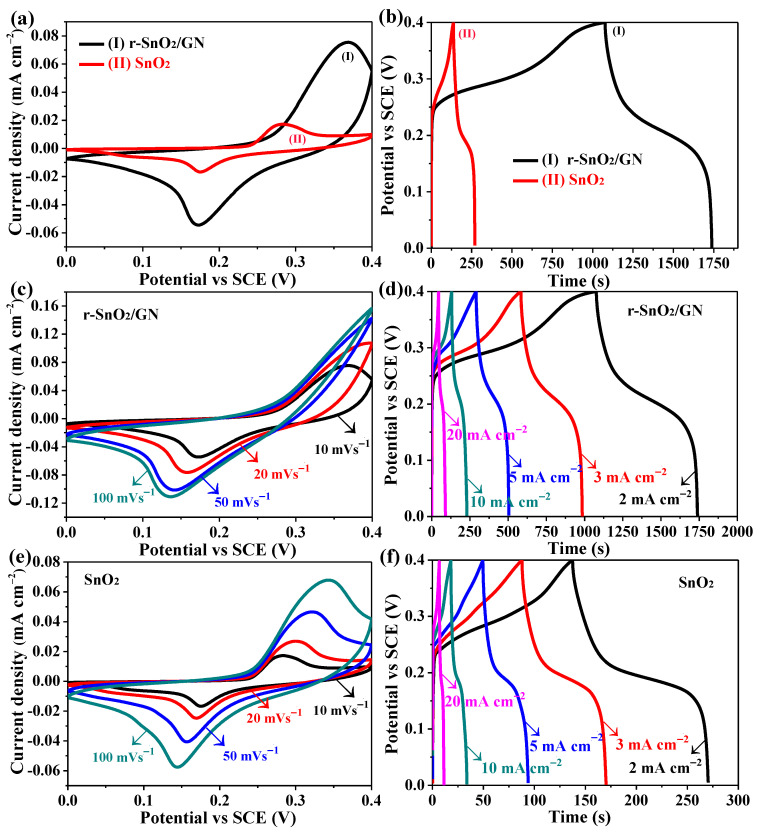
(**a**) CV curves of two electrodes measured at a scan rate of 50 mVs^−1^; (**b**) GCD curves of two electrodes tested at a current density of 2 mA cm^−2^; CV curves for electrodes (**c**) r-SnO_2_/GN and (**e**) SnO_2_ measured at various voltage sweep rates; GCD curves for electrodes (**d**) r-SnO_2_/GN and (**f**) SnO_2_ measured at various current densities.

**Figure 7 nanomaterials-11-00372-f007:**
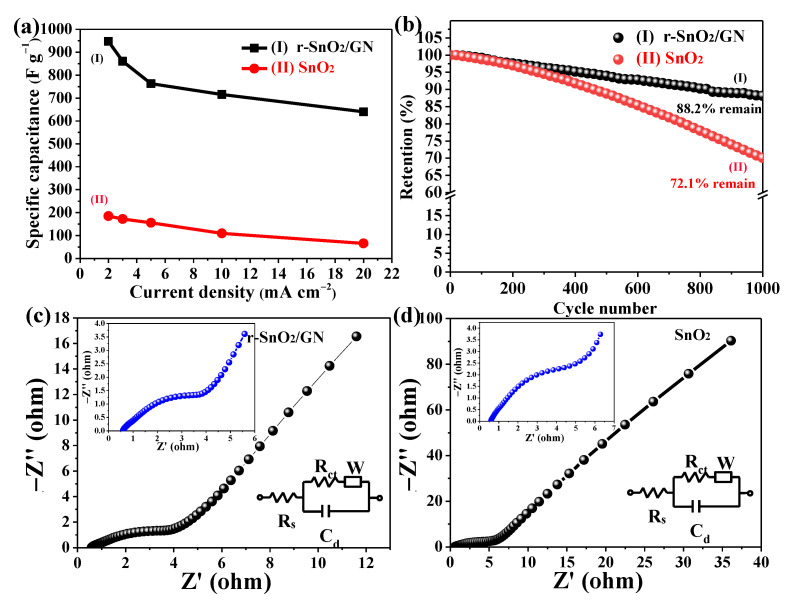
(**a**) Specific capacitance as a function of current density; (**b**) The cycle stability; EIS Nyquist plots and the equivalent circuit model (the inset) for electrodes (**c**) r-SnO_2_/GN and (**d**) SnO_2_.

**Figure 8 nanomaterials-11-00372-f008:**
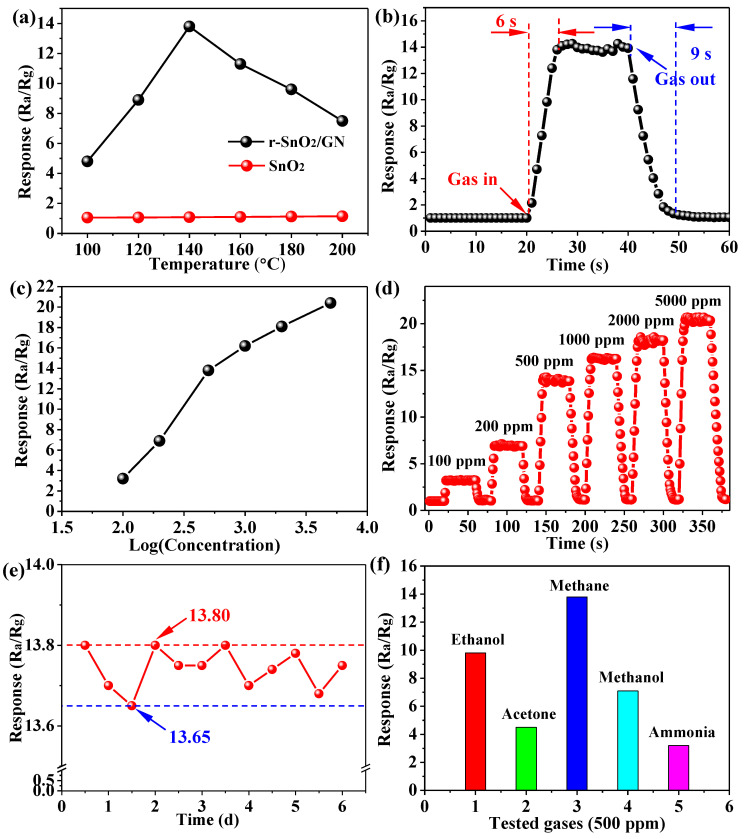
(**a**) Two gas sensors toward methane gas of 500 ppm at different working temperatures; (**b**) A particular response curve of the sensor r-SnO_2_/GN toward methane gas at 140 °C; (**c**) The response value of the sensor r-SnO_2_/GN toward methane gas vs. log(concentration, ppm) and (**d**) the corresponding response curve; (**e**) The stability of the sensor r-SnO_2_/GN toward methane gas at 140 °C; (**f**) The response value of the sensor r-SnO_2_/GN toward various gases.

**Table 1 nanomaterials-11-00372-t001:** The brief summary of electrochemical performance of SnO_2_-based supercapacitors.

Electrode Materials	Electrolyte	Current Density	Specific Capacitance (F g^−1^)	Ref.
SnO_2_	0.5 M Na_2_SO_4_	1 A g^−1^	138	[34]
SnO_2_-NGO	6.0 M KOH	4 A g^−1^	378	[35]
SnO_2_/g-C_3_N_4_	1.0 M Na_2_SO_4_	1 A g^−1^	488	[47]
Hollow SnO_2_	1.0 M KOH	1 A g^−1^	332.7	[30]
r-SnO_2_-GN	3.0 M KOH	2 mA cm^−2^ (0.57 A g^−1^)	947.4	This work

**Table 2 nanomaterials-11-00372-t002:** The brief summary of sensing performance of SnO_2_-based CH_4_ sensors.

Sensing Materials	Temperature (°C)	CH_4_ Concentration	Reponses	Ref.
Pt-SnO_2_	400	1000 ppm	1.55 ^b^	[36]
Pd-SnO_2_	400	6600 ppm	20 ^b^	[37]
Pt-SnO_2_	350	1000 ppm	4.5 ^b^	[49]
SnO_2_-rGO	150	1000 ppm	47.6% ^a^	[50]
Pd–SnO_2_	340	3000 ppm	17.6 ^b^	[51]
r-SnO_2_-GN	140	1000 ppm 5000 ppm	16.2 ^b^ 20.4 ^b^	This work

^a^*S* = (Δ*R*/*R*_g_) × 100%. ^b^
*S* = *R*_a_/*R*_g_ × 100%.

## Data Availability

No new data were created or analyzed in this study. Data sharing is not applicable to this article.

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
