# Peer review of "Oxygen-Deficient Stannic Oxide/Graphene for Ultrahigh-Performance Supercapacitors and Gas Sensors"

_nanomaterials, 2021, doi:10.3390/nano11020372_

Round 1

Reviewer 1 Report

This work reports the synthesis of a composite material constituted of oxygen-deficient SnO2 sub-micrometer particles and reduced graphene oxide (rGO). There are two proposed applications for these materials: supercapacitor electrode and gas-sensing (particularly methane). The concept for a SnO2 / rGO nanocomposite has already been proposed in the literature for electrochemical applications in several published research papers. In order to reach the publication stage, the authors should address many important issues.

Specifically, the most important are:

  • The novelty element in the synthesis method should be clearly highlighted in the manuscript in relation with the existing literature
  • The authors claim that the obtained SnO2 particles are oxygen-deficient; the main proof for it comes from a slight shift in binding energy of the Sn 3d doublet in the photoelectron spectrum. However, only one reference is provided ([38]). Since this is quite a strong claim, it is suggested to provide more references, specifically pertaining oxygen-vacancies in SnO2, if possible. Moreover, it would be interesting to provide also the XPS characterization of the SnO2 material without rGO.
  • The authors should include in the discussion of results a comparison of performance as supercapacitor electrode and gas-sensor with the existing literature on SnO2-based materials

In addition, the following improvements are suggested:

  • The resolution of figure 2 and figure 3 is low and it is hard to appreciate the electron microscopy images
  • It would be interesting to have BET and XPS results for the SnO2 material without rGO
  • In figure 5, the deconvolution of the O 1s region provides no peak identification of the components

Author Response

We appreciate these nice comments from the reviewer. Please see the attachment.

Reviewer 2 Report

Author has investigated on the synthesis of Oxygen-Deficient Stannic Oxide/Graphene for Ultrahigh-Performance Supercapacitors and Gas Sensors application. I would like to suggest few comments to author to consider before its publication.

  1. Raman analysis of both samples should be included in the revised manuscript.
  2. It would be interesting to see the elemental analysis (EDS spectroscopy and or elemental mapping) of both samples.
  3. The cycling stability of supercapacitor electrodes in Fig. 7b seems to be around 88% at 1000 cycles and what would be the stability after 2000 cycles?
  4. The stability of r-SnO2/GN nanocomposites towards gas sensor (Fig. 8) should be included.

Author Response

(The authors gave the same response as above.)

Reviewer 3 Report

The authors report SnO2/graphene nanocomposites for supercapacitor and gas sensor applications. The results are clear that the nanocomposite has vividly enhanced performance compared with SnO2. I do recommend the publication of the manuscript in the journal. However, there are several issues that should be clarified prior to publication. 1. Raman or other similar spectroscopy is needed to confirm the existence of graphene in the composite. 2. For supercapacitors, there are many studies on metal oxide/graphene composites reported. Please compared the performance of the material with them. Table might be needed. 3. How did you optimize the content of graphene for the composite? 4. For the sensor application, base resistance of the materials is important. Resistances of SnO2 and the nanocomposite should be compared. Are both materials n-type? Since graphene is p-type , the nanocomposite can show p-type conduction depending on the content of it. 5. The gas response vs concentration plot can be changed to show better linearity: response vs (concentration)^1/2 (or log(concentration) or log (response) vs log (concentration)

Author Response

(The authors gave the same response as above.)

Round 2

Reviewer 1 Report

The authors have improved the manuscript following the suggestions by the reviewer; therefore, this work deserves publication in the present form.